# Anti-inflammatory cytokine profile and Jarisch-Herxheimer reaction in Leptospirosis patients: A prospective case-series study in New Caledonia

Julie Cagliero[1,2], Anne Loarec[1]*, Julien Lebon[3], François Baur[4], Patrick Lefevre[5], Emilie Follenfant[3], Cyrille Goarant[1,6], Damaris Ukeiwe[1], Julien Colot[1,3], Sylvie Tardieu[7], Catherine Werts[2]*‡, Cécile Cazorla[3]‡

1 Institut Pasteur de Nouvelle Calédonie, Médipole, Dumbéa, Nouvelle-Calédonie, France, 2 Institut Pasteur Paris, Unité de Biologie et génétique de la paroi bactérienne, Paris, France, 3 Centre Hospitalier Territorial Gaston Bourret, Médipole, Dumbéa, Nouvelle-Calédonie, France, 4 Centre Hospitalier Territorial du Nord, Poindimié, Nouvelle-Calédonie, France, 5 Centre Hospitalier Territorial du Nord, Koumac, Nouvelle-Calédonie, France, 6 The Pacific Community, Public Health Division, Nouméa, Nouvelle-Calédonie, France, 7 Centre Hospitalier Territorial du Nord, Koné, Nouvelle-Calédonie, France

‡ These authors are joint senior authors on this work and co-last authors.
* cwerts@pasteur.fr (CW); aloarec@pasteur.nc (AL)

## Abstract

### Background

Leptospirosis is a neglected zoonosis. This spirochetal disease is common in tropical countries where rainfall and poor sanitation facilitate skin contact with environmental *Leptospira* shed in animal urine. Antibiotics are effective against spirochetes, although a harmful Jarisch-Herxheimer (JHR) reaction can occur within hours of treatment, with the onset of chills, fever and/or hypotension. However, the awareness and incidence of JHR in leptospirosis are poorly understood.

### Methods

This prospective observational study enrolled 81 patients diagnosed with leptospirosis from four hospitals in New Caledonia between 2021 and 2024. To evaluate the patients' inflammatory status and identify risk factors for JHR, we collected data on clinical, socioeconomic, and biological factors (including blood cytokine levels) at admission and during the hours following treatment with different regimens of β-lactam antibiotics.

### Main results

The majority of the cohort were middle-aged men, most of them Melanesian farmers. They exhibited high levels of C-reactive protein (CRP), neutrophilia, thrombocytopenia, and elevated biochemical markers indicative of liver and kidney dysfunction, which are typical

**Data availability statement:** Institut Pasteur is committed to sharing and disseminating health data from its programs and research in an open, timely, and transparent manner to promote health benefits for populations while respecting ethical and legal obligations toward patients, research participants, and their communities. However, the LEPJAR-NC study faces specific ethical and regulatory constraints. The population of New Caledonia is small (approximately 270,000 people), and the number of hospitals involved in this research is limited. Furthermore, although leptospirosis is an important public health concern, it remains a relatively low-incidence disease. Taken together, these factors create a risk that even de-identified datasets may permit the re-identification of individual patients, particularly when combined with other public or local health records. Our data remain accessible in a transparent and responsible manner. A full dataset description, variable dictionary, and access request procedure have been deposited on a public OWEY platform (https://doi.org/10.48802/owey.yY8sZyxp). The pseudonymized dataset is accessible to qualified researchers upon request, pending a review that complies with our legal and ethical commitments. Interested parties may contact the LEPJAR-NC Data Access committee at dac-lepjar-nc@pasteur.fr to initiate this process.

**Funding:** The author(s) received no specific funding for this work.

**Competing interests:** The authors have declared that no competing interests exist.

of leptospirosis. Unexpectedly, pro-inflammatory cytokine levels were low or undetectable upon admission, while high levels of the anti-inflammatory cytokine IL-10 were measured. After antibiotherapy, increased levels of the pro-inflammatory cytokines TNF and IL-6, as well as IL-10 were observed. Strikingly, there was no increase in IL-1ß, the main player in the "cytokine storm". JHR, identified with a new clinical score, occurred in 48% (possibly 61%) of patients and was associated with higher cytokine levels, as expected.

## Conclusion/Significance

This study confirms the stealth nature of leptospires, which induce a potent anti-inflammatory response rather than inflammation. It calls into question both the cytokine storm hypothesis, which is often cited in leptospirosis and the use of immunosuppressive drugs. The high incidence of JHR in New Caledonia suggests that the systematic use of ß-lactams as a first-line treatment should be reevaluated

## Author summary

Leptospirosis is a neglected zoonosis caused by spirochetal bacteria. Antibiotics are effective against leptospires, but they can induce an adverse reaction, called the Jarish-Herxheimer reaction (JHR), which occurs in the hours following the initiation of treatment. The present study was conducted in several hospitals in New Caledonia to measure the JHR incidence and describe the cytokine evolution. We included 81 patients diagnosed with leptospirosis and treated with antibiotics. A very interesting result was the absence of inflammation in the blood of patients before antibiotic treatment, confirming that leptospires are stealth bacteria not responsible for inflammation. In contrast, antibiotic treatment triggered JHR in about 50 percent of patients and was associated with increased inflammation, but not with markers of leptospirosis severity, nor cytokine storm. This study may raise the alarm on JHR and question the use of anti-inflammatory drugs and the choice of antibiotics in leptospirosis.

## Introduction

Leptospirosis is a neglected re-emerging zoonotic disease with worldwide distribution caused by pathogenic spirochetes bacteria of the genus *Leptospira*. More than sixty species and three hundred serovars have been described so far, but only a small group of highly virulent strains, including *L. interrogans,* are pathogenic to humans and animals [1]. The physiopathology of leptospirosis remains poorly understood. The disease is transmitted by penetration through the abraded skin or mucosa of bacteria present in the soil or water, initially shed into the environment in urines of chronically infected animals [2]. Pathogenic leptospires then reach the bloodstream and disseminate in target organs such as kidneys, lungs and liver.

Leptospirosis is a public health challenge. In 2015, the global incidence of human leptospirosis was estimated worldwide at 1 million cases and nearly 60,000 deaths per year, making it one of the most threatening bacterial zoonoses worldwide, with an incidence of 3 to 150/100,000 per year [3] and a fatality range from 5 to 30%, depending on disease awareness and health systems [4,5]. The prevalence of leptospirosis is high in tropical and subtropical areas, and in low-income countries with poor sanitation, where seasonal rains facilitate the spread of leptospires through contaminated sewage or flooding. In addition to the economic burden of livestock morbidity, this disease affects the most vulnerable populations in emerging countries, with a higher burden than cholera in terms of disability-adjusted life-years of approximatively 3 million days lost per year, mainly among men [5]. Vaccines based on inactivated *Leptospira* are available in a few countries (mainland France, Cuba, China), but provide only short-lasting and serovar-specific protection. Despite its health and economic burden, leptospirosis is not listed as a neglected disease by the World Health Organization (WHO) [6–8]. Nevertheless, leptospirosis is one of the notifiable diseases to the health authorities in France and in New Caledonia.

Most cases of human leptospirosis are asymptomatic or may be mistaken for other tropical febrile syndromes, such as influenza, dengue or hemorrhagic fevers. About 10% of cases progress to severe forms, including Weil's disease with high bacteremia, hemorrhage, and multi-organ failure. The host immune response to infection is thought to be an important factor in the development of leptospirosis, as are the infecting strain and the bacteremia [9,10]. Cytokines, particularly TNF-α, IL-6, IL-1β and IL-10, have been found to be higher in severe and even fatal forms of human leptospirosis compared to milder infections [10,11]. It is hypothesized that imbalance in the innate immune response and a possible subsequent cytokine storm phenomenon may be associated with leptospiral progression to severe syndromes [12]. However, inflammation in leptospirosis remains unclear. Indeed, studies in mouse models show that leptospires largely evade the immune system, especially phagocytes. Moreover, although they induce cytokine production *in vitro*, they induce little or even inhibit inflammation *in vivo*, especially when compared to inflammation induced by other bacteria, such as *Escherichia coli* [13,14]. To date, there are no precise, robust data on the inflammatory state of human patients at the time of suspected leptospirosis, particularly before the antibiotic treatment is initiated or when the disease progresses to severe forms.

WHO guidelines recommend an antibiotic treatment, treating severe cases with high dose of intravenous penicillin and less severe cases with oral antibiotics such as amoxicillin, ampicillin, doxycycline or erythromycin [4]. Azithromycin is also used in some countries, with a slightly better tolerance compared to doxycycline [15–17].WHO guidelines recommend an antibiotic treatment, treating severe cases with high dose of intravenous penicillin and less severe cases with oral antibiotics such as amoxicillin, ampicillin, doxycycline or erythromycin [4]. Azithromycin is also used in some countries, with a slight better tolerance compared to doxycycline [15–17]. Third-generation cephalosporins, such as ceftriaxone and cefotaxime, and quinolone also appear to be effective [4]. To date, no acquired antibiotic resistance has been reported in leptospires. A delay in the administration of antibiotics has been associated with severe forms of leptospirosis, justifying their use even though their beneficial effects have been questioned after meta-analysis of randomized clinical trials following standard Cochrane procedures. In fact, there appears to be no beneficial effect of antibiotics on lethality when administered to patients with late or severe leptospirosis [15,16].

Antibiotic treatment of spirochetal infections can trigger a febrile inflammatory syndrome, the Jarisch-Herxheimer Reaction (JHR), which can significantly worsen the patient's condition in the hours following the initiation of antimicrobial therapy [18]. JHR, first described in patients with syphilis treated with mercury, usually occurs in the first 12 hours of antimicrobial treatment in patients with spirochetal infections, with a peak at 6–8 hours [18]. The clinical presentation of a JHR remains poorly characterized but a profound deterioration in patient condition with inflammatory symptoms (especially high fever, headaches, shivering or rigors, and decreased blood pressure) is usually reported. This syndrome is described as transient and self-limited. The underlying pathophysiology of the JHR is not understood and this syndrome has long been mistaken for penicillin allergy or worsening of the disease. Studies suggest that antibiotics may lead to the release of spirochetes cell wall components. These immunogenic and pro-inflammatory molecules may indeed induce a massive innate immune response, which may explain the increase in cytokines and chemokines levels observed during a JHR in

patients with relapsing fevers due to *Borrelia* species, especially TNF-α, IL-6, and IL-8, as well as the associated symptoms, such as chills, sweating and tachycardia [18].

The JHR phenomenon is much less documented in leptospirosis infection than in other spirochetal infections. A systematic review of 28 published studies worldwide estimated the occurrence of JHR to be 9% of 976 leptospirosis patients treated with antibiotics [19]. However, the incidence of JHR seems to be very variable in different countries. For example, JHR was estimated at 44% in the Futuna Islands, a French territory in the Pacific ocean [20].

The French island of New Caledonia, located in the South Pacific Ocean, presents a variety of geographic, ethnic and socioeconomic characteristics. The northern part of the island is humid with a low population density, mostly Melanesian communities with a rural and tribal lifestyle, at risk of leptospirosis. The population seeks medical care at the Centre Hospitalier du Nord (CHN) in Koné or at one of its two branches in Koumac or Poindimié. The south of the island and the urban area of Nouméa gathers 75% of the island's population, of various ethnic origins, who have easier access to the Centre Hospitalier Territorial Gaston Bourret (CHT). Some practitioners follow WHO guidelines by administering a full dose of antibiotics, mainly penicillin or cephalosporins, when severe leptospirosis is suspected. Given the limited access to intensive care in some parts of the island, several medical care centers have developed a therapeutic protocol in which patients suspected of having leptospirosis receive escalating doses of amoxicillin over the first 6 hours of treatment until the full dose is reached, aiming to prevent the dreaded development of JHR. Finally, in the remote northern part of New Caledonia, steroidal anti-inflammatory drugs may be associated with antimicrobial protocols. A retrospective study of patients with confirmed leptospirosis estimated the incidence of JHR between 2005 and 2007 to be 12% in the north of New Caledonia [20]. The authors suggest that JHR may be associated with cardiovascular failure and aggravation of renal injury, necessitating close monitoring in the first hours after antibiotic initiation. However, the influence of these treatments on the outcome of leptospirosis as well as the occurrence of JHR remains to be studied. One issue in the fields of leptospirosis and syphilis is the lack of a precise definition or scoring system for JHR, that would allow for easier comparison between studies [18,19,21–23].

This project is an observational prospective study aimed at collecting clinical and biological data to determine the inflammatory status of patients confirmed with leptospirosis on admission and during the hours following the initiation of the antibiotherapy. A description of the affected population, treatments and outcomes, and a new clinical JHR scoring were performed to better understand and determine the risk factors and incidence of JHR in New Caledonia.

## Materials and methods

### Ethics statement

The LEPJAR-NC study was approved by the Comité de protection des personnes d'Ile de France III (avis CPP #3897-RM), by the Comité Consultatif d'Ethique de Nouvelle-Calédonie (Avis #2020–11001) and by the comité d'éthique du Centre Hospitalier Territorial (CHT) de Nouvelle-Calédonie (avis CNRIPH # 21.02.17.63715). All patients included in the study were given an information note and provided free and informed verbal consent to the physician who treated them at the time of suspected leptospirosis.

### Study settings

The study was conducted at four hospitals in New Caledonia: the CHT (the only tertiary-level hospital in New Caledonia with intensive care unit), located near Nouméa in the southwest; and the three branches of the CHN in the north: CHN Koné (a secondary-level hospital), and CHN Koumac and CHN Poindimié (both primary-level hospitals).

### Inclusion of patients

Patients aged 18 years and older attending the hospital emergency departments for suspected leptospirosis were consecutively recruited during the rainy epidemic season between December 2021 and May 2024. Free and informed consent

was obtained before enrollment. A socio-demographic questionnaire was administered during hospitalization. Patients were excluded from the study if they had a chronic inflammatory disease or were receiving concomitant antibiotic (ATB) and/or anti-inflammatory treatment or medical management that was incompatible with the aim of the study. Finally, participants were excluded from the study if the diagnosis of leptospirosis was not confirmed. No sample size was defined; all patients suspected with leptospirosis during the study period were included. All emergency department staff were trained in the study procedures.

## Study design

After usual care, patients were placed under medical supervision for 24 hours (S1 Fig). Before the start of antibiotics, a baseline assessment (H0) was performed, including clinical examination and blood sampling, which was repeated at hour 3 (H3) and hour 6 (H6) after the start of the treatment. The patients' clinical condition was reevaluated 24 hours after treatment. The evolution of pre-existing symptoms and the appearance of new clinical signs were recorded at each time point, with particular attention paid to the evolution of hemodynamic status and febrile symptoms. If the condition of the study participants required admission to an intensive care unit, no additional samples were collected for the study. When possible, biological analyses were performed on the remaining blood samples collected during conventional medical management.

## Biological analyses

Various analyses were performed on blood samples collected from study participants at different time points (25 ml at H0, and 15 mL at H3 and H6, when the patient's condition permitted). The blood samples from all sites were transported to CHT laboratory for routine hematology and biochemistry analysis. C-reactive protein (CRP) was measured without waiting for confirmation of the patient's diagnosis. Leptospirosis diagnostic and genotyping was performed at the Institut Pasteur of New Caledonia, as well as the cytokine assays for the patients confirmed with leptospirosis. Pro-inflammatory cytokines (TNF-α, IL-1β, IL-6) and anti-inflammatory cytokine IL-10 were quantified from sera using Duoset kits ELISA (R&D System) according to the manufacturer's instructions. Concentrations were calculated using Excel software. Quality controls have been performed according to the manufacturer recommendation to ensure the validity of the laboratory results in human sera.

## Diagnosis of leptospirosis

Patients were diagnosed with leptospirosis if they tested positive for *Leptospira* by qPCR in blood or urine at the time of enrollment [24], or if they had detectable levels of IgM antibodies to *Leptospira* species in serum, as determined by a commercial IgM ELISA (Panbio Pty Ltd, Queensland, Australia). Quantification of bacteriemia were performed at CHT by real-time PCR against the *lipL32* gene, conserved in pathogenic P1 *Leptospira* species [24], using MagNA Pure 96 (Roche diagnostics) et QIA Symphony Qiagen DNA purification kits. After leptospirosis DNA diagnosis, eluates were stored at -20°C for further identification of species and genogroups by qPCR against the *lfb-1* gene and direct sequencing of amplicons [25]. Serogroup and serovars have not been determined by the micro-agglutination test (MAT) but previous correspondence between serogroups and genotypes has been established [26].

## Treatments of patients

The administration of antibiotics at different sites followed various protocols. Full-dose protocols involved administering ß-lactams antibiotics; amoxicillin (Clamoxyl 1 g, three times a day) or third-generation cephalosporins (3CGs), such as Cefotaxime (1–2 g, three times a day) or Ceftriaxone (1–2 g per day), via direct intravenous injection (IVD). The latter was sometimes preceded by 1 mg/kg of methylprednisolone sodium succinate (Solumedrol, IVD). In Koumac, patients were treated with a progressive escalating amoxicillin protocol together with Hydrocortisone. Treatment began at H0 with

intravenous Hydrocortisone (50 mg) plus oral Clamoxyl (25 mg). Then, Clamoxyl was administered orally every hour at the following doses: H1: 50 mg, H2: 125 mg, H3: 250 mg, and H4: 500 mg. At H5, Clamoxyl (1 g) was administered intravenously along with Hydrocortisone (50 mg). This was repeated every six hours for 24 hours.

## Classification of JHR

JHR was defined as follows during the first 6 hours of treatment: onset or recurrence of fever (defined as a temperature ≥38°C) or onset of chills or myalgia were considered as 2 major criteria, while increased heart rate above 20%, increased respiratory rate above 20%, and hypoxaemia (defined as a O2 saturation below 95% or decision to start oxygen therapy) were considered as 3 minor criteria (Table 1). Blood pressure variation was not included because it could have been confused with a worsening of leptospirosis. The criteria evolution was defined compared to the baseline assessment (H0) before antibiotherapy and if happening within the first 6 hours after treatment initiation. In fact, JHR symptoms occur suddenly within hours following antibiotic administration and resolved rapidly—typically within 6–12 hours—suggesting a transient, treatment-associated reaction rather than the natural course of the infection, whereas deterioration of patients may include other symptoms such as the typical renal and hepatic failure. Patients with 2 major criteria were classified as "JHR", those with 1 major and 1 minor or 2 minor criteria as "probable JHR", and those with 1 criterion or none as "no JHR". If data were missing, the JHR category was "not defined" (ND) (Table 2). Clinicians reviewed all cases and clinical records, and the final JHR status was reported taking into account the clinician's opinion. The distinction between JHR and disease progression was based on the clinician's clinical interpretation, considering the timing, symptom dynamics, and overall patient trajectory.

## Data collection

The medical team collected individual data, using standardized forms. After anonymization, data clerks entered the files into a secure database with Research Electronic Data Capture (REDCap) software [27] using a password-protected account at the Institut Pasteur, Paris. Baseline patient characteristics were described: demographic characteristics (sex,

**Table 1. New JHR score: major and minor criteria according to the clinical status, LEPJAR-NC study, New Caledonia, 2021–2024.**

| Major criteria | Minor Criteria |
|---|---|
| Onset or recurrence of fever (ear temperature °C ≥ 38°C) | Heart rate increase (increase of more or equal than 20% between the start of treatment and hour 6) |
| Onset or recurrence of chills or myalgias (reported by patients themselves) | Respiratory rate increase (increase of more or equal than 20% between the start of treatment and hour 6) |
| | O$_2$ saturation decrease under 95% or decision to start oxygen therapy |

**Table 2. categorization of the JHR status, LEPJAR-NC study, New Caledonia, 2021–2024.**

| JHR Status | | | |
|---|---|---|---|
| **Yes** | **Probable** | **No** | **ND** |
| 2 major criteria | 1 major + 1 minor OR 2 minor + missing information | 1 minor or no criteria | Missing data |

age, self-reported community), clinical and biological characteristics (temperature, hemodynamic parameters, whole blood count, liver and kidney function, CRP, cytokines), comorbidities and risk factors (occupational and recreational activities), as well as some key variables at follow up H3, H6 and for some patients 24h post-treatment. Case definitions and categories for characteristics were based on international recommendations and scientific evidence.

## Statistical analysis

Categorical variables were summarized using percentages and compared using the Pearson χ2 test or Fisher's exact test. Continuous variables were summarized using median and interquartile ranges (IQR) and compared using the Kruskal-Wallis Test (Non-Parametric ANOVA) as appropriate. The distribution of continuous variables was assessed using the Shapiro-Wilk test. For the analysis of the evolution of cytokines over time, a Wilcoxon signed-rank test for paired data (non-parametric) was performed. A two-sided p-value less than 0.05 was considered statistically significant.

We first tested the hypothesis that socio-demographic and biological characteristics of patients at admission differed between sites. Associations between JHR status and baseline characteristics were also assessed. Finally, we tested the hypothesis that the evolution of cytokine levels between H0 and H3, and between H3 and H6, differed according to JHR status. Patients with unknown status were not considered in the analysis. Missing information was not included in the analysis; the denominators reflect the number of informative observations. We performed data analysis using R software (4.4.1 version).

## Results

### General description

From December 2021 to May 2024, a total of 194 patients were enrolled; out of them, 81 patients (41.75%) were confirmed with leptospirosis (Fig 1), n = 75 by positive PCR in blood, n = 4 in urine, and n = 2 by IgM ELISA. The CHT included 29 patients (35.80%). Fifty-two patients (64.20%) were enrolled in one of the 3 sites of the CHN: 33 (40.74%) patients in Koumac, 10 (12.35%) in Koné, and 9 (11.11%) in Poindimié. One patient was enrolled in 2021, 54 in 2022, 21 in 2023 and 5 in 2024 during the seasonal outbreaks.

The median age of patients with leptospirosis was 44 years (IQR: 31.75,56.00) and 59 patients (72.84%) were male (Table 3). Fifty-five patients (67.90%) self-identified as belonging to the Melanesian, i.e. Kanak, community. Out of 66 patients with information, nine patients (13.64%) were affected by one or more comorbidities: hypertension (n = 7), diabetes (n = 5), gout (n = 2), history of cerebral ischemic stroke (n = 1), heart failure (n = 1) (Table 3). There was no evidence of a difference between hospital sites for gender, comorbidities and community affiliation. However, there were significant differences in occupation (p value: 0.016) and freshwater related activities (p value: 0.0006) between the different hospital sites. Indeed, 3 out of 4 patients admitted in Poindimié (75%) and 16 out of 23 patients (70%) admitted in Koumac were farmers, but only 5 out of 24 patients (20%) admitted in the CHT. Also, the proportion of leptospirosis patients with freshwater related activities varied from 88.9% (8/9) in the patients admitted in Poindimié to 34.48% (10/29) in the patients admitted in the CHT. There was no evidence of a difference in animal-related activities and field occupation (Table 3).

The figure details treatment regimens (amoxicillin in full dose or in progressive administration, third-generation cephalosporins [3GCs], with or without hydrocortisone, or 3GCs in combination with gentamycin or nitro-imidazole), and subsequent classification of patients according to JHR events, possible JHR, no JHR, or missing JHR information. Percentages are given relative to the number of patients within each treatment category.

### Baseline assessment

Patients' blood was analyzed on admission for leptospiral load, for white and red blood cell counts and characteristics, renal and hepatic markers, C-Reactive protein (CRP), and cytokines levels. For the 74 patients with positive PCR in blood, the median bacteriemia was calculated to be 1,136 leptospires/mL (Table 4). Genotyping was performed in 67 patients; *L.*

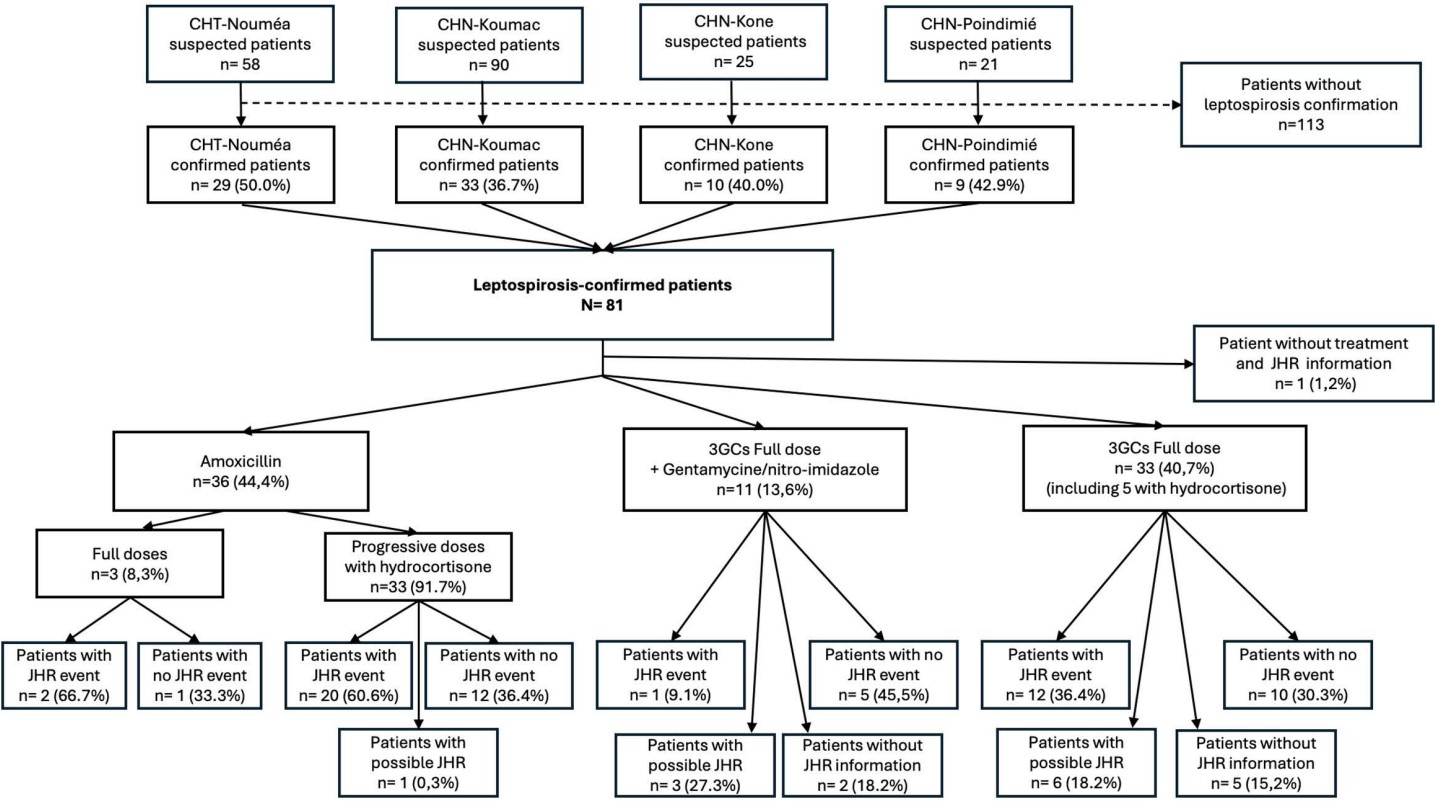

**Fig 1. Flow diagram of patient inclusion, antibiotic treatment, and occurrence of Jarisch–Herxheimer reaction (JHR) among 81 laboratory-confirmed leptospirosis cases in New Caledonia.**

*interrogans* was found in 88% of patients, including 40 (60%) with genotype I1, corresponding to the Icterohaemorragiae serovar, previously evidenced as a major risk of severe leptospirosis (26), 10 (15%) with genotype I2 (serovar Australis), 9 (13%) with genotype I5 (Pyrogenes), and *L. borgpeterseni* genotype B1 (Ballum) was found in 8 patients (12%). The distribution of genotypes among sites was different (p-value: 0.0023) (Table 4). There was no significant differences in sociodemographic characteristics or risk factors across genotypes, except for exposure to river water: all patients infected with genotype I5 reported such exposure (8/8), compared to 4/7 for B1, 18/36 for I1, and 7/10 for I2. The distribution of genotypes also differed significantly by hospital of admission (p-value:0.00231), likely to reflect differences in environmental exposures in the corresponding catchment areas (S3 Table). No statistical difference of bacteriema was found between the different genotypes (S4 Table).

Hyperleukocytosis was observed with a median of 11,640 cells/µl, mainly due to neutrophilia (10,580 neutrophils/µl). Thrombocytopenia was measured at 71,500 platelets/µL. Hemoglobin, hematocrit and urea levels were normal. Elevated median values of bilirubin (27.4 UI/mL), gamma-glutamyl transferase (gamma-GT) (67 UI/mL), aspartate (ASAT) (67.5 UI/mL) and alanine (ALAT) (55 UI/mL) transaminases levels were suggestive of hepatic dysfunction. In addition, the median of creatinine, a marker of renal function was also elevated (129 µmol/L). Of note, platelet count, ASAT, ALAT and creatinine were statistically different between the sites, and potentially linked to the genotype difference (S4 Table). Regarding inflammatory markers, the median CRP was above the normal with a marked elevation at 235 mg/mL, but the pro-inflammatory cytokines, IL-1ß and IL-6 were undetectable in most of the patients. Very low levels of TNF were detected with a median of 45 pg/mL, while very high levels of the anti-inflammatory cytokine IL-10 were detected with a median of

**Table 3. Socio-demographic description and risk factors of the enrolled patients, LEPJAR-NC study, New Caledonia, December 2021–Mai 2024.**

| | Total N=81 n/N (%) or median [IQR] | CHT N=29 n/N (%) or median [IQR] | Koumac N=33 n/N (%) or median [IQR] | Koné N=10 n/N (%) or median [IQR] | Poindimié N=9 n/N (%) or median [IQR] | Verbal value* |
|---|---|---|---|---|---|---|
| **Age (years)** n=80 | 44 [31.75-56.00] | 43 [30.00-59.00] | 45 [31.50-56.00] | 44 [36.25-58.25] | 48 [42.00-55.00] | 0.748 |
| **Sex** Male | 59/81 (72,84) | 25/29 (86.21%) | 23/33 (69.70%) | 6/10 (60.00%) | 5/9 (55.56%) | 0.145 |
| **Community group** | | | | | | |
| Melanesians | 55/81 (67.90%) | 17/29 (58.62%) | 21/33 (63.64%) | 9/10 (90.00%) | 8/9 (88.89%) | 0.844 |
| Europeans | 7/81 (8.64%) | 3/29 (10.34%) | 4/33 (12.12%) | 0/10 (0.00%) | 0/9 (0.00%) | |
| Other Communities† | 6/81 (7.41%) | 4/29 (13.79%) | 2/33 (6.06%) | 0/10 (0.00%) | 0/9 (0.00%) | |
| Not declared/ Missing | 13/81 (16.05%) | 5/29 (17.24%) | 6/33 (18.18%) | 1/10 (10.00%) | 1/9 (11.11%) | |
| **Comorbidities** Yes | 9/66 (13.64%) | 6/27 (22.22%) | 1/21 (4.76%) | 1/9 (11.11%) | 1/9 (11.11%) | 0,346 |
| **Occupation** | | | | | | |
| Farmer | 24/53 (45.28%) | 5/24 (20.83%) | 16/23 (69.57%) | 0/2 (0.00%) | 3/4 (75,00%) | **0,016** |
| Out of work | 13/53 (24.53%) | 9/24 (37.50%) | 4/23 (17.39%) | 0/2 (0.00%) | 0/4 (0.0%) | |
| Workman | 11/53 (20.75%) | 6/24 (25.00%) | 2/23 (8.70%) | 2/2 (100%) | 1/4 (25,00%) | |
| Self-employed | 2/53 (3.77%) | 2/24 (8.33%) | 0/23 (0.00%) | 0/2 (0.00%) | 0/4 (0.00%) | |
| Student | 2/53 (3.77%) | 1/24 (4.17%) | 1/23 (4.35%) | 0/2 (0.00%) | 0/4 (0.00%) | |
| Manager | 1/53 (1.89%) | 1/24 (4.17%) | 0/23 (0.00%) | 0/2 (0.00%) | 0/4 (0.00%) | |
| **Animal-related activities** Yes | 62/75 (82.67%) | 21/29 (72.41%) | 24/ 28 (85.71%) | 9/9 (100%) | 8/9 (88.89%) | 0,272 |
| **Field occupation** Yes | 50/ 75 (66.67%) | 18/29 (62.07%) | 21/ 28 (75.00%) | 3/9 (33.3%) | 8/9 (88.89%) | 0,062 |
| **freshwater related activities** Yes | 47/75 (62.67%) | 10/29 (34.48%) | 23/ 28 (82.14%) | 6/9 (66.67%) | 8/9 (88.89%) | **0,0006** |

Acronyms: CHT: Centre Hospitalier Territorial, IQR: Inter-quartile Range.

* Fisher's exact test/ Kruskal-Wallis Test (Non-Parametric ANOVA) to assess differences between the enrollment sites.

†Other communities include Asian, Polynesian, Indonesian, Vietnamese, and people from Vanuatu.

1,477 pg/mL, one thousand time more than the norm in healthy adults. No statistically significant difference was observed for CRP and cytokines between the different sites (Table 4).

**Treatment regimen**

Patients were treated by clinicians as per their usual practice (Fig 1). Treatment information was not recorded for one patient. All patients (80) were treated with beta-lactams, either third-generation cephalosporins (3GCs) or amoxicillin, although treatment regimen differed according to site of care (S1 Table). All patients with 3GCs (n=44/80, 55.00%) were treated according to the WHO recommended "full dose" protocol. In addition to the 3GCs, ten patients (12.50%) received gentamicin, an aminoside, and one patient (1.25%) received gentamicin with nitroimidazole, two antibiotics from classes other than ß-lactams. Thirty-six patients (44.4%) were treated with amoxicillin, most of whom received a progressive protocol (91.7%). The administered antibiotic dose was increased during the first 6 hours of treatment until the full dose was reached. Of note, the progressive protocol with amoxicillin, used at CHN-Koumac, was systematically associated with the concomitant administration of corticosteroidal anti-inflammatory drug, hydrocortisone. A total of thirty-three patients (40.7%) were treated with escalating doses of amoxicillin and thirty-three patients (40.7%) were treated with full dose 3GCs alone, including five treated with Methylprednisolone. A total of thirty-eight patients (46.91%) received corticosteroids (Fig 1).

**Table 4. Description of the biological results at admission of the enrolled patients with confirmed leptospirosis, LEPJAR-NC study, New Caledonia, 2021–2024.**

| | Total<br>N=81<br>n/N (%) or<br>Median [IQR] | CHT<br>N=29<br>n/N (%) or<br>Median [IQR] | Koumac<br>N=33<br>n/N (%) or<br>Median [IQR] | Koné<br>N=10<br>n/N (%) or<br>Median [IQR] | Poindimié<br>N=9<br>n/N (%) or<br>Median [IQR] | p-value* |
|---|---|---|---|---|---|---|
| **Bacteraemia** (Leptospires/mL), n=74 | 1,136<br>[110-11,628] | 1,865<br>[109-14,951] | 545<br>[85-4,962] | 1,681<br>[339-3,028] | 2,113<br>[214-11,988] | 0,6998 |
| **Genotype**<br>*L. interrogans* I1 | 40/67 (59.70%) | 20/24 (83.33%) | 9/25 (36.00%) | 8/10 (80.00%) | 3/8 (37.50%) | **0,0023** |
| *L.interrogans* I2 | 10/67 (14.93%) | 0/24 (0.00%) | 5/25 (20.00%) | 1/10 (10.00%) | 4/8 (50.00%) | |
| *L.interrogans* I5 | 9/67 (13.43%) | 1/24 (4.17%) | 6/25 (24.00%) | 1/10 (10.00%) | 1/8 (12.50%) | |
| *L. borgpeterseni* B1 | 8/67 (11.94%) | 3/24 (12.50%) | 5/25 (20.00%) | 0/10 (0.00%) | 0/8 (0.00%) | |
| **Blood Cells**†<br>Leucocytes (/µl), n=79 | **11,640**<br>[8,500-13,990] | **11,750**<br>[9,500-15,030] | 10,000<br>[8,300-12,800] | **14,350**<br>[1,150-18,575] | 8,800<br>[7,300-12,900] | 0,1213 |
| Lymphocytes (/µl), n=76 | **560**<br>[400-900] | **510**<br>[350-760] | **600**<br>[400-1, 000] | **800**<br>[650-850] | 500<br>[450-660] | 0,4901 |
| Neutrophiles (/µl), n=77 | **10,580**<br>[7,100-13,160] | **10,950**<br>[8,510-13,680] | **8,800**<br>[7,000-110-11,600] | **13,100**<br>[9,775-16,600] | **11,200**<br>[7,450-12,564] | 0,2675 |
| Monocytes (/µl), n=70 | 500<br>[300-762] | 490<br>[270-770] | 400<br>[300-800] | 700<br>[650-900] | 500<br>[360-600] | 0,5316 |
| Thrombocytes/µL, n=78 | **71,500**<br>[35,000-118,500] | **63,000**<br>[32,000-107,000] | **87,000**<br>[52,000-149,000] | **43,500**<br>[15,250-71,000] | **113000**<br>[34,750-131,500] | **0,0267** |
| **Red blood cells**<br>Haematocrit (%), n=70 | 39.00<br>[35.78-41.15] | 39.00<br>[35.00-41.00] | 38.00<br>[35.00-41.25] | 39.60<br>[38.80-40.40] | 40.00<br>[39.75-41.50] | 0,6297 |
| Haemoglobin (g/dl), n=74 | 13.70<br>[12.03-15.00] | 13.00<br>[12.00-14.00] | 13.65<br>[12.10-15.00] | 13.00<br>[12.00-14.00] | 14.00<br>[13.70-15.50] | 0,3134 |
| **Kidney function**‡<br>Urea (mmol/L), n=75 | 8<br>[5 -18] | **10**<br>[6–21] | 7<br>[5–11] | 18<br>[11–22] | 12<br>[5–16] | 0,1795 |
| Creatinine (µmol/L), n=79 | **129**<br>[101-263] | 244<br>[113-383] | 107<br>[87-176] | 192<br>[122-318] | 123<br>[104-160] | **0,0149** |
| **Hepatic function**<br>ASAT (UI/L), n=74 | **67**<br>[39-137] | 126<br>[67-240] | 56<br>[35 -98] | 92<br>[51-152] | 37<br>[3,353] | **0,0013** |
| ALAT (UI/L), n=75 | 55<br>[30-104] | **77**<br>[44-130] | 53<br>[23-68] | **81**<br>[40-122] | 31<br>[26-48] | **0,0417** |
| Gamma-GT (UI/L), n=39 | **67**<br>[32-114] | **85**<br>[32-140] | **67**<br>[65-72] | 55<br>[27-105] | 21<br>[21–21] | 0,3952 |
| Bilirubin (µmol/L), n=73 | **27**<br>[19 -97] | **62**<br>[20-135] | **23**<br>[16-38] | **85**<br>[33-190] | **23**<br>[20–34] | 0,1985 |
| **C-Reactive Protein** (mg/L), n=75 | **235**<br>[177-295] | **272**<br>[209-324] | **226**<br>[177-288] | **198**<br>[83-258] | **200**<br>[185-200] | 0,0986 |
| **Cytokines** (pg/mL)§<br>TNF, n=71 | **45**<br>[18-81] | **47**<br>[8-122] | **39**<br>[22-69] | **77**<br>[40-108] | **54**<br>[13-99] | 0,5410 |
| IL-10, n=72 | **1,477**<br>[533-2,947] | 1,991<br>[813-3,415] | 1,266<br>[383-2,655] | 1,824<br>[773-3,185] | 1,391<br>[548 -2,634] | 0,7124 |
| IL-1ß, n=72 | 0<br>[0-0] | 0<br>[0-0] | 0<br>[0-0] | 0<br>[0-0] | 0.00<br>[0-0] | 0,4873 |
| IL-6, n=71 | 0<br>[0-51] | 0<br>[0-82] | 0<br>[0-0] | 0<br>[0-252] | 0<br>[0-0] | 0,2963 |

*Fisher's exact test/ Kruskal-Wallis Test (Non-Parametric ANOVA) to assess differences between enrollment sites.

†Normal blood cells value in adult healthy population, source CHT, Nouméa: Leucocytes: 4 – 10 G/L, Lymphocytes: 1–5 G/L, Neutrophiles: 2 - 7.5 G/L, Monocytes: 0.2 – 1 G/L, Thrombocytes 150–400 G/L (150 000–400 000/µl), Hematocrit: women: 38–47% and men 40–54%, Hemoglobin: women 12–16g/dl and men 13–17g/dl;

‡Biochemical parameter in adult healthy population, source: CHT, Nouméa: urea: 2.5 to 9.2 mmol/L (depending on age and sex), creatinine: 49–104 µmol/L (depending on age and sex), ASAT: 5–34UI/L, ALAT: 0–55 UI/L, G-GT: <38 for women and <55 for men, bilirubin total: 5.1-20.5 µmol/L; CRP<5mg/l.

§Cytokine levels in healthy humans [28]; TNF<30 pg/mL, IL-10<17 pg/mL, IL-1ß not detected, IL-6<15 pg/mL For each parameter, n represents the total number of patients tested. Values in bold are outside the norm, or, for p values, indicate a significant difference between sites.

## Evolution following antibiotherapy

Two types of parameters were followed at H3 and H6 after starting the treatment: the clinical evaluation with hemodynamic status and the inflammatory markers; CRP and cytokines.

### Clinical evolution

Of the 81 patients, clinical data were not collected for two patients, and H6 data were missing for one patient. Movement in intensive care units (ICU) was recorded for only 11 patients. The transfer to the ICU was associated with the severity of the patients' clinical and biological presentations upon admission. Patients transferred from North health facilities were admitted directly to the CHT ICU. This transfer may have also been motivated by minimal surveillance capacity in the north. No additional information concerning ICU transfers (e.g., time, clinical and biological assessments, or care) was collected. Only clinical signs associated with suspected JHR were recorded on the case report form. Of the 59 patients with information, 40 (67.80%) presented an increase or recurrence of fever and 33 of 42 patients with data (78.57%) presented chills. Regarding the respiratory and hemodynamic parameters collected, 31/58 presented an increase in heart rate, 20/35, an increase in respiratory rate, and 18/58 a decrease in oxygen saturation. A general assessment was also recorded at H24, except for 7 patients. Of the 81 patients, 37 patients were identified with a worsening of their clinical status, mostly due to acute respiratory distress syndrome, hemodynamic disturbances, or deterioration of renal function. Two patients died during this observation period, both with acute respiratory distress syndrome. One of them was co-infected with COVID-19.

### Pro/ anti-inflammatory markers

The levels of CRP measured at baseline (H0) and also at H3 and H6 remained comparable and were not statistically different as assessed by the Wilcoxon signed-rank test for paired data (non-parametric) (Table 5).

Of the 81 patients enrolled, 62 patients had complete follow-up according to the study protocol with cytokine levels at the 3 time points. Some results at H3 and H6 were not available. The median of TNF concentration was 122 pg/mL at H3 and 124 pg/mL at H6 compared to 45 pg/mL at H0. Using a Wilcoxon signed-rank test for paired data, there was a statistically significant difference between the TNF level at H0 and at H3 ($p < 0.001$), which was increased, but not between H3 and H6. The median of IL-6 level increased from undetectable at H0 to 57 pg/mL at H3 and to 862 pg/mL at H6. Statistical evidence of difference levels between H0 and H3, and H3 and H6 was observed ($p < 0.001$ for both). The concentration of IL-10 showed evidence of an increase with a median at H3 of 2,738 pg/mL compared to 1,474 pg/mL at H0 ($p < 0.001$), and to 5,183 pg/mL at H6 ($p < 0.01$). No such increase was observed for IL-1ß levels, which remained below the detection limit with a median of 0 pg/mL at H0, H3 and H6 (Table 5).

In conclusion, the proinflammatory cytokines TNF and IL-6, which were barely detectable at inclusion, significantly increased after antibiotic treatment at H3 and were still elevated at H6. Notably, IL-1ß did not increase after antibiotic treatment. In contrast, the anti-inflammatory cytokine IL-10, which was already very high at inclusion, further increased significantly after antibiotic treatment.

### Classification of JHR

In the absence of a predefined protocol for the classification of leptospirosis patients, the diagnosis of JHR was made according to a new scoring system, inspired by a previous study conducted in New Caledonia [20], and literature review. This new scoring was developed with the clinicians of the different study sites to define and harmonize the classification of JHR in leptospirosis (Tables 1 and 2). Eight patients did not have the information concerning the JHR status and were excluded for the following analysis. Overall, among the 73 patients, 35 presented a JHR (48%), 10 a probable JHR (13, 6%) and 28 (38%) a no JHR status (Fig 1).

**Table 5. Evolution of the CRP and the cytokines between H0 (admission), H3 and H6, LEPJAR-NC study, New Caledonia, 2021–2024.**

| | H0 Median [IQR] | H3 Median [IQR] | H6 Median [IQR] | Diff H3-H0 p-value* | Diff H6-H3 p-value* | Diff H6-H0 p-value |
|---|---|---|---|---|---|---|
| **CRP** (mg/L) | 235.00 n=75 [177.00-295.30] | 243.70 n=61 [178.10-282.40] | 231.40 n=55 [175.90-286.50] | 0.9045 | 0.07827 | 0.415 |
| **TNF** (pg/mL) | 45.00 n=71 [18.50-81.00] | 122.00 n=66 [29.25-310.75] | 124.00 n=63 [30.00-348.50] | **<0.001** | 0.4672 | **<0.001** |
| **IL-10** (pg/mL) | 1,477.00 n=71 [533.75-2947.00] | 2738.00 n=67 [670.50-13224.00] | 5,183.00 n=63 [1013.50-19572.00] | **<0.001** | **0.0058** | **<0.001** |
| **IL-1ß** (pg/mL) | 0.00 n=72 [0.00-0.00] | 0.00 n=67 [0.00-0.00] | 0.00 n=63 [0.00-0.00] | 1 | 1 | 1 |
| **IL-6** (pg/mL) | 0.00 n=71 [0.00-51.50] | 57.00 n=66 [0.00-5312.75] | 862.00 n=62 [0.00-7670.00] | **<0.001** | **0.0093** | **<0.001** |

*Wilcoxon signed-rank test for paired data (non-parametric).

## Risk factors for JHR Status

All centers identified patients with JHR, but the proportions varied between the sites (p-value<0.001). Among the patients with JHR score, 60.6% (20/33) of the patients enrolled at Koumac were classified as presenting a JHR, 50.0% (3/6) at Poindimié, 35.7% (10/28) at the CHT, and 33.3% (2/6) at Kone (Table 6), The socio-demographic characteristics of the patients were explored to assess differences according to the JHR status, but no difference was found (Table 6). Among patients classified with JHR, 17 out of 35 (48.6%) experienced clinical worsening at H24. This proportion was 5 out of 10 (50%) for patients with possible JHR, and 10 out of 28 (35.7%) for patients without JHR. We were unable to demonstrate a statistical evidence (p=0.268).

## Treatment received and JHR status

Due to confounding effects between groups and small sample sizes, we were unable to reliably assess the impact of treatment on JHR occurrence (Fig 1). The comparison between treatment protocols (escalating versus full dose) was impaired by the adjunction or not of cortoicoids, and by the type of ß-lactam antibiotic used (amoxicillin or 3GCs, respectively). Nevertheless, the group that received the escalating-dose regimen of amoxicillin and hydrocortisone had a higher proportion of "JHR" and a lower proportion of "probable JHR" classifications than those who received a full-dose regimen. This suggests different levels of awareness and expertise regarding JHR among the sites (S2 Table).

## Biological parameters and JHR status

We assessed if biological indicators were different at admission for patients who later developed a JHR and if the evolution of the cytokines differed according to the JHR status. The patients who developed a JHR after treatment initiation presented a median bacteremia of 2,179 leptospires/mL, compared to 4,046 for the patients developing a probable JHR and 338 for the patients who didn't develop a JHR (Table 7). A difference between the three groups was also found concerning the lymphocytes with a p-value at 0.0454.

Unexpectedly, the patients who developed a JHR had a median of liver and renal parameters lower than the one with no JHR. A statistically significant difference between the 3 groups and between "JHR" and "no JHR" was found for urea, creatinine, ASAT and bilirubin. To note that this difference was significant for ALAT only between "JHR" and "no JHR". Concerning the inflammatory parameters at inclusion, there was no significant difference for the CRP between the three groups. On the same way, the TNF levels, the IL-10 and IL-1ß did not show a significant difference; a significant difference was only found for the cytokine IL-6, with a median at admission at 0 for the patients "JHR" and "no JHR", and 74 for the probable JHR.

**Table 6. Description of the socio-demographic parameters according to JHR, LEPJAR-NC study, New Caledonia, 2021–2024.**

| | Total n/N (%) | JHR n/N (%) or median [IQR] | Probable JHR n/N (%) or median [IQR] | No JHR n/N (%) or median [IQR] | p-value* | p-value† (JHR/ No JHR) |
|---|---|---|---|---|---|---|
| **Hospital** CHT (missing = 1) | 28 (100) | 10 (35.71) | 5 (17.86) | 13 (46.43) | **<0.001** | 0.435 |
| Koumac | 33 (100) | 20 (60.61) | 1 (3.03) | 12 (36.36) | | |
| Koné (missing = 4) | 6 (100) | 2 (33.33) | 2 (33.33) | 2 (33.33) | | |
| Poindimié (missing = 3) | 6 (100) | 3 (50.00) | 2 (33.33) | 1 (16.67) | | |
| **Age** (year) | NA | 46.00 [27.50-55.50] | 42.00 [34.25-63.00] | 44.00 [33.00-52.50] | 0,799 | 0.887 |
| **Sexe** Female | 18 (100) | 11 (61.11) | 2 (11.11) | 5 (27.78) | 0.278 | 0.257 |
| Male | 55 (100) | 24 (43.64) | 8 (14.55) | 23 (41.82) | | |
| **Community** Melanesians | 49 (100) | 24 (48.98) | 7 (14.29) | 18 (36.73) | 0.971 | 0.764 |
| Europeans | 7 (100) | 4 (57.14) | 1 (14.29) | 2 (28.57) | | |
| Other Communities‡ | 5 (100) | 3 (60.00) | 0 (0.00) | 2 (40.00) | | |
| Missing/ not declared | 12 (100) | 4 (33.33) | 2 (16.67) | 6 (50.00) | | |
| **Comorbidities** No | 51 (100) | 23 (45.10) | 7 (13.73) | 21 (41.18) | 0.760 | 0.674 |
| Yes | 8 (100) | 4 (50.00) | 2 (25.00) | 2 (25.00) | | |
| **Occupation** Farmer | 23 (100) | 12 (52.17) | 2 (8.70) | 9 (39.13) | 0.936 | 0.673 |
| Out of work | 12 (100) | 6 (50.00) | 1 (8.33) | 5 (41.67) | | |
| Workman | 11 (100) | 6 (54.55) | 2 (18.18) | 3 (27.27) | | |
| Self-employed | 2 (100) | 0 (0.00) | 0 (0.00) | 2 (100) | | |
| Student | 2 (100) | 1 (50.00) | 0 (0.00) | 1 (50.00) | | |
| Manager | 1 (100) | 1 (100) | 0 (0.00) | 0 (0.00) | | |
| **Animals linked activities** No | 12 (100) | 10 (83.33) | 0 (0.00) | 2 (16.67) | 0.064 | **0.0497** |
| Yes | 56 (100) | 23 (41.07) | 9 (16.07) | 24 (42.86) | | |
| **Field occupation** No | 24 (100) | 13 (54.17) | 3 (12.50) | 8 (33.33) | 0.662 | 0.588 |
| Yes | 44 (100) | 20 (45.45) | 6 (13.64) | 18 (40.91) | | |
| **Freshwater related activities** No | 26 (100) | 13 (50.00) | 4 (15.38) | 9 (34.62) | 0.902 | 0.790 |
| Yes | 42 (100) | 20 (47.62) | 5 (11.90) | 17 (40.48) | | |

Acronyms: CHT: Centre Hospitalier Territorial, IQR: Inter-quartile Range.

*Fisher's exact test/ Kruskal-Wallis Test (Non-Parametric ANOVA) to assess differences between the JHR statuses.

†Exclusion of the patients with a status "JHR possible" Fisher's exact test/ Kruskal-Wallis Test (Non-Parametric ANOVA) to assess differences between the JHR Status.

‡Other communities include Asian, Polynesian, Indonesian, Vietnamese and people from Vanuatu.

### Evolution of inflammatory parameters and JHR status

The CRP levels did not show major changes between H0 and H6, for any JHR status group.

Compared to H0, a significant increase in TNF was observed at H3 in patients classified as "JHR" and in patients classified as "probable JHR" (Table 8). Increased levels of IL-6 and IL-10 cytokines were also observed at both H3 and H6 in most of the "JHR" and "probable JHR" groups. In contrast, no increase in IL-10, TNF and IL-6 was observed in the "no JHR" group. No increase in IL-1ß was observed in any of the patients. Of note, 8 patients with severe symptoms and presenting the highest cytokine levels were equally distributed between the "JHR" and "no JHR" groups.

**Table 7. Description of the biological parameters according to JHR status, LEPJAR-NC study, New Caledonia, 2021-2024.**

| | JHR n/N (%) or Median [IQR] | Probable JHR n/N (%) or Median [IQR] | No JHR n/N (%) or Median [IQR] | p-value* | p-value[†] (JHR/ No JHR) |
|---|---|---|---|---|---|
| **Leptospiral load** (Leptospires/mL) | 2,179 [154 -14,212] | 4,046 [1,242-25,817] | 338 [79-1,244] | **0,016** | 0,051 |
| **Genotype** L. borgpeterseni B1 | 6 (75.00%) | 0 (0.00%) | 2 (25.00%) | 0,579 | 0.344 |
| L. interrogans I1 | 15 (41.67%) | 6 (16.67%) | 15 (41.67%) | | |
| L. interrogans I2 | 5 (62.50%) | 1 (12.50%) | 2 (25.00%) | | |
| L. interrogans I5 | 5 (62.50%) | 2 (25.00%) | 1 (12.50%) | | |
| **Leucocytes** (/µl) | 11,500 [8,650-13, 000] | 9,500 [8,800-13,400] | 11,425 [8,125-17,390] | 0,861 | 0,623 |
| **Lymphocytes** (/µl) | 500.00 [400-700] | 500 [400-800] | 700 [505-1,200] | **0,045** | **0,015** |
| **Neutrophiles** (/µl) | 9,400 [7,160-11,825] | 8,900 [6,800-11,670] | 9,300 [6,850-14,455] | 0,704 | 0,447 |
| **Monocytes** (/µl) | 470 [300-770] | 280 [250-460] | 600.00 [400-845] | 0,109 | 0,363 |
| **Thrombocytes** (/µl) | 97,000 [40,000-152, 000] | 47,000 [26,750-104,250] | 71,500 [49,000-105,250] | 0,206 | 0,213 |
| **Hematocrit** (%) | 40 [37–43] | 39 [34–40] | 37 [35–41] | 0,173 | 0,063 |
| **Hemoblobin** (g/dl) | 14 [12–15] | 13 [11–14] | 13.00 [12–14] | 0,478 | 0,548 |
| **Urea** (mmol/L) | 6.95 [5–9] | 6.00 [6 –13] | 14.50 [6-26] | **0,001** | **0,002** |
| **Creatinine** (µmol/L), | 107 [89-145] | 128.00 [113-131] | 229 [105-397] | **0,0211** | **0,007** |
| **ASAT** (UI/L), | 58 [34-99] | 93 [57-177] | 104 [57-172] | **0,0207** | **0,011** |
| **ALAT** (UI/L), | 44 [23-74] | 67 [57-109] | 68 [38 -134] | 0,061 | **0,037** |
| **Gamma-GT** (UI/L), | 53 [25-94] | 40.50 [34-61] | 89 [47-139] | 0,240 | 0,121 |
| **Bilirubin** (µmol/L), | 23 [15-34] | 30 [19-70] | 84 [21-226] | **0,006** | **0,001** |
| **CRP** (mg/L), | 236 [200-297] | 213 [136-246] | 195 [158-304] | 0,156 | 0,226 |
| **TNF** (pg/mL) | 32 [13-70] | 74 [30-99] | 43 [17-110] | 0,450 | 0,547 |
| **IL-10** (pg/mL) | 2,219 [524-3,043] | 2,091[1,141-3,057] | 940 [213-2,657] | 0,196 | 0,138 |
| **IL-1ß** (pg/mL) | 0 [0-0] | 0 [0 -0] | 0 [0-0] | 0,733 | 0,462 |
| **IL-6** (pg/mL) | 0 [0-0] | 74 [17-917] | 0 [0-1] | **0,005** | **0,666** |

*Fisher's exact test/ Kruskal-Wallis Test (Non-Parametric ANOVA) to assess differences between the JHR statuses.

[†]Exclusion of the patients with a status "probable JHR" Fisher's exact test/ Kruskal-Wallis Test (Non-Parametric ANOVA) to assess differences between the JHR statuses.

In conclusion, this prospective study with 81 leptospirosis patients from New Caledonia showed that a large proportion of the cohort was middle aged men, most of them Melanesian farmers or with freshwater-related occupations. They exhibited the usual biological perturbations associated with leptospirosis, such as high CRP levels, neutrophilia, thrombocytopenia and elevated biochemical markers suggestive of liver and kidney dysfunction. Interestingly, despite the presence of bacteria and neutrophils in blood, levels of pro-inflammatory cytokines were very low or undetectable, whereas high levels of the anti-inflammatory IL-10 were measured, confirming the stealthiness of leptospires. Second, this observational study showed that JHR, established with a new score, occurred in 48% (possibly 61%) of patients treated in different hospitals of New Caledonia with various regimen of ß-lactam antibiotics. Interestingly, JHR was associated with increased cytokine levels, although no increased in IL-1ß was observed, dismissing the "cytokine storm" hypothesis. A tendency of higher leptospiral loads at inclusion, as well as statistically significant lower levels of renal and hepatic dysfunctions markers was observed in patients who developed a JHR upon antimicrobial therapy.

**Table 8. Median of difference between H3 and H0, and H6 and H3 for the different inflammatory parameters according to JHR status, LEPJAR-NC study, New Caledonia, 2021–2024.**

| | Total obs | JHR | Probable JHR | No JHR | p-value* | p-value† |
|---|---|---|---|---|---|---|
| | | Median [IQR] | Median [IQR] | Median [IQR] | | (JHR/ No JHR) |
| CRP (H3-H0) (mg/L) | 55 | 5.75 [-18; 46] | 10.80 [-0; 198] | -14.20 [-36; 6.27] | **0.0338** | 0.0625 |
| CRP (H6-H3) (mg/L) | 52 | -13.35 [-34; 7] | 2.05 [-31;15] | -4.45 [-7; 0] | 0.5560 | 0.2782 |
| TNF (H3-H0) (pg/mL) | 62 | **104 [22; 240]** | **307 [9; 612]** | **0 [-26; 50]** | **0.0026** | **0.0018** |
| TNF (H6-H3) (pg/mL) | 58 | -8. [-51; 67] | -15 [-251;24] | 0 [-9;29] | 0.5945 | 0.7098 |
| IL-10 (H3-H0) (pg/mL) | 63 | **5,279 [642; 15,584]** | **5,417 [-81; 16,605]** | **0 [-387; 475]** | **0.0039** | **0.0006** |
| IL-10 (H6-H3) (pg/mL) | 58 | 3,204 [-1,321; 10,956] | 861 [-1,076; 10,186] | 0 [-161; 4,239] | 0.5914 | 0.3567 |
| IL-1ß (H3-H0) (pg/mL) | 63 | 0 [0;0] | 0 [0;0] | 0 [0;0] | 0.8519 | 0.7786 |
| IL-1ß (H6-H3) (pg/mL) | 58 | 0 [0;0] | 0 [0;0] | 0 [0;0] | 0.6750 | 0.8539 |
| IL-6 (H3-H0) (pg/mL) | 62 | **525 [0; 6,115]** | **1535 [200; 8,623]** | **3 [0; 521]** | 0.2639 | 0.1970 |
| IL-6 (H6-H3) (pg/mL) | 57 | **672 [-187; 8,100]** | **56 [-5,200; 3,740]** | **5 [0; 451]** | 0.8768 | 0.6776 |

*Fisher's exact test/ Kruskal-Wallis Test (Non-Parametric ANOVA) to assess differences between the JHR statuses.

†Exclusion of the patients with a status "JHR probable" Fisher's exact test/ Kruskal-Wallis Test (Non-Parametric ANOVA) to assess differences between the JHR statuses.

## Discussion

We aimed in this prospective study to better understand the occurrence of JHR in patients with leptospirosis in New Caledonia. Indeed, JHR is an overlooked reaction in the field of leptospirosis. Some clinicians are not aware of it, whereas others use it as an early diagnostic tool to discriminate leptospirosis from other diseases [20]. We took advantage of high prevalence of leptospirosis in New Caledonia to characterize inflammation in patients' blood before and in the hours following antibiotic administration. To the best of our knowledge this is new since clinical studies including cytokines dosage in leptospirosis patients are usually done after enrollment and antibiotic treatment [11,29–31]. Importantly, the present results show that leptospirosis causes anti-inflammatory but almost no pro-inflammatory cytokine response in hospitalized patients which is consistent with our previous data obtained in lethal mouse model of leptospirosis [14]. This total absence of inflammation in patients with leptospirosis was unexpected because of the elevated levels of CRP, considered as an inflammatory marker of leptospirosis and bacterial infections that correlate with IL-6 [32]. However, inflammation occurred in patients with leptospirosis after antibiotic treatment, suggesting, as already hypothesized for JHR, that the ß-lactam antibiotics targeting the cell wall of leptospires lead to the release of fragments recognized by the innate immune system [18]. Interestingly, the present study confirmed the association between JHR, defined from clinical signs, and inflammation. *Ex vivo* studies with human blood from healthy volunteers are ongoing to understand the mechanism of JHR and decipher which are the leptospiral components released and what are the host receptors involved. The present study suggests that the JHR may be involved in the misleading perception of leptospirosis as an inflammatory disease that may lead to cytokine storm and multi-organ failure [12]. Notably, although TNF and IL-6 cytokines were increased after antibiotic treatment of patients with leptospirosis, no secretion of IL-1ß was found in this study. In comparison, a 2015 study measured IL-1β levels in the sera of 47 patients with acute leptospirosis at a mean of 40 pg/mL. However, this level was not significantly different from that of healthy controls [30]. Similar IL-1β levels were found in a 2013 Brazilian study, though leptospirosis patients exhibited higher levels of IL-1β than controls. However, there was no difference between mild and severe patients [33]. These results differ greatly from those of studies on sepsis. For example, in a 2016 study of patients with acute cystitis, the mean IL-1β level was 500 pg/mL, whereas control groups exhibited undetectable levels of IL-1β [34]. The lack of IL-1β secretion in patients with leptospirosis is an important finding because IL-1ß is a central orchestrator of inflammation, and is a marker for many bacterial infections that cause sepsis. In fact, IL-1ß has paracrine

and amplifying effects and with TNF can lead to uncontrolled inflammation and the so-called cytokine storm [35]. This absence of circulating IL-1ß has also been shown in a mouse model [14], consistent with the fact that leptospires block the lytic cell death called pyroptosis, thereby drastically limiting the secretion of IL-1ß [13]. Furthermore, the JHR is known to be transient, with a self-resolution within 24 hours [18] which is inconsistent with the cytokine storm theory. In fact, although we did not record the cytokines 24 h after treatment, TNF did not increase between H3 and H6, and together with the absence of IL1ß, rejects the cytokine storm hypothesis.

Whether JHR could aggravate and worsen the patient condition, sometimes leading to organ failure is an interesting question. However, in the previous study of JHR in New Caledonia, it was found that renal failure and thrombocytopenia, markers of severe leptospirosis were not associated with JHR [20]. The present study confirms that JHR was not associated with severe leptospirosis, at least at admission, since patients with JHR had lower levels of leptospirosis severity markers such as ASAT, creatinine and urea at admission. The latter was also found to be lower in patients with JHR in the previous study [20]. In addition, although 37/81 (45%) of patients showed worsening of their symptoms, and 48% were classified as JHR, we observed as many patients with severe leptospirosis and highest levels of cytokines in the "JHR" than in the "no JHR" groups. Nevertheless, to conclude about whether JHR exacerbates leptospirosis symptoms, as is sometimes suggested with a worsening of renal, hepatic, platelet and other markers, biological data would have needed to be collected 24 hours after the start of antibiotherapy. Unfortunately this data was missing for most patients.

Notably, although nearly 60% of patients from this cohort were infected with *L. interrogans* serovar Icterohaemorrhagiae, which has been linked to severe leptospirosis in New Caledonia [26], Koumac patients had the highest incidence of JHR, but were only 30% to be infected with this serovar. On the other hand the Australis serovar, which was previously shown to be a risk factor for JHR in the North of New Caledonia, was found in 30% of patients in Koumac. This suggests that inside the same species here, *L. interrogans*, the leptospiral serovar may affect the JHR.

The previous study conducted between 2007 and 2009 in New Caledonia and Futuna Island suggested that 44% of patients with leptospirosis in Futuna and 12% in New Caledonia presented JHR [20]. Here, we found a marked higher level of JHR, around 50%. This was unexpected since at least in Koumac, the clinician and population did not change between the two studies. In addition, the cohorts in other centers (Poindimié, Koné and CHT) did not differ, with young active people, and a large majority of Melanesian men, often with a professional or personal activity linked to the environment. This discrepancy may be due to the difference in JHR diagnostic. Indeed, although the same parameters (shivering and rise in temperature as major criteria and >20% change in hemodynamic and respiratory parameters as secondary signs) were used in both studies, these criteria were not equally pondered. Another hypothesis is the difference in leptospirosis diagnostic. Here, 91% of enrolled patients (74/81) were PCR positive in blood indicating the presence of circulating *Leptospira*, whereas in the previous study, only 61% of patients were PCR positive and/or were MAT positive, which means that the latter had been already sick enough time to get antibodies. It may also suggest, in parallel with the mouse model [36,37], that less or no more leptospires could be present in the bloodstream of the MAT positive patients, with potentially less release of leptospiral components in blood upon antibiotic treatment. In this line, in the present study, bacterial load was positively associated with the occurrence of JHR after antibiotic treatment, with the JHR and "probable JHR" groups with higher bacterial loads at inclusion compared to "no JHR" group. Altogether these data suggest that patients who will develop JHR are those with leptospires still in the blood at the time of treatment, which corresponds to an early phase of infection, before the deterioration of renal and hepatic function that may occur. This is consistent with a short delay between disease onset and antibiotic treatment, which was found to be a risk factor for JHR in the previous study [20].

This study has several limitations that reduce its scope and significance. First, the delay between onset of first leptospirosis symptoms and antibiotic treatment was not recorded in the present study. It would have been an important parameter to consider confirming the above hypothesis. In addition, because the study was observational and prospective, it was not possible to correlate the JHR with a specific treatment. In fact, the different treatments and management of patients

with leptospirosis according to the different regions and hospital facilities allowed only a few analyses. The possible correlation between JHR and the use of amoxicillin versus 3CGs would have been interesting since some countries like Thailand using 3CGs or doxycycline (7) do not observe JHR, but it was not possible due to the confounding use of escalating and corticoids protocols. Nevertheless, the higher occurrence of JHR in Koumac does not support for the escalating protocol using amoxicillin combined with hydrocortisone. However, in Koumac, almost all patients were classified as JHR and only one was in the "probable JHR" group; based on the clinician's expertise, whereas in other sites, 8 patients (almost 20%) treated with full dose and no anti-inflammatory drugs were classified as "possible JHR" and 5 could not be classified. It is therefore possible that the proportion of patients with JHR was underestimated, but that the data collected, and the opinion of the clinicians did not allow us to classify them with certainty. In Koné and CHT, many visiting doctors come from mainland France. They are not necessarily aware of leptospirosis, and are even less aware of JHR. However, it is important to note that that a high percentage of patients were included in the "JHR and Probable" groups at all sites (63.03% in Koumac, 53.57% in CHT, 66.66% in Poindimié, and 83.33% in Koné). Despite various antibiotic regimen, all of these patients received ß-lactam antibiotics. This suggests that alternative treatments used in other countries, such as doxycycline or azithromycin, should be considered, as JHR does not seem to be an issue there.

The protocols with corticosteroids were designed to help reduce the JHR. Methylprednisone was used in Poindimié, but in a hybrid form of management with a full dose 3GCs protocol. Similarly, no conclusions could be drawn about the potential efficacy of the progressive protocol and hydrocortisone to reduce inflammation. The beneficial effect of glucocorticoids in leptospirosis is not clearly established. A study in Iranian patients with leptospirosis treated with ceftriaxone, a ß-lactam antibiotic, with or without prednisolone, showed no beneficial effect on the evolution of leptospirosis except for a shorter recovery from the thrombocytopenia, a hallmark of leptospirosis [38]. In contrast, in the case of acute lung injury due to severe leptospirosis, the early administration of corticosteroids reduced the mortality [39]. However, the present results showing the lack of inflammation on admission, demonstrate that leptospires by themselves do not induce inflammation in humans, and the absence of IL-1ß after antibiotic treatment showed that the antibiotic treatment for leptospirosis did not either induce a cytokine storm. Therefore, our study did not suggest any benefit of limiting inflammation in patients with leptospirosis.

## Supporting information

**S1 Fig. Flow of the study protocol for inclusion and data and sample collection, LEPJAR-NC study, New Caledonia, 2021–2024.**
(DOCX)

**S1 Table. Description of the leptospirosis treatment of the enrolled patients, LEPJAR-NC Study, New Caledonia, 2021–2024.**
(DOCX)

**S2 Table. Description of the treatment regimen received according to JHR status, LepjarNC Study, New Caledonia, 2021–2024.**
(DOCX)

**S3 Table. Description of the socio-economic factors of enrolled patients according to the leptospiral genotype, LEPJAR-NC Study, New Caledonia, 2021–2024.**
(DOCX)

**S4 Table. Description of biological data of enrolled patients according to leptospiral genotype, LEPJAR-NC Study, New Caledonia, 2021–2024.**
(DOCX)

# Acknowledgments

We would like to thank Roman Thibeaux for retrieving and granting us access to the data, and reading the manuscript. We would like to expres our gratitude to the patients who made this study possible. We would like to thank the CHT and CHN hospital staff, in particular the state- nurses (IDE).

All authors have read and approved the final version of the manuscript. The corresponding authors had full access to all data in this study and takes complete responsibility for the integrity of the data and the accuracy of the data analysis.

## Author contributions

**Conceptualization:** Julie Cagliero, Patrick Lefevre, Cyrille Goarant, Catherine Werts, Cécile Cazorla.

**Data curation:** Anne LOAREC, Emilie Follenfant.

**Formal analysis:** Julie Cagliero, Anne LOAREC.

**Investigation:** Julie Cagliero, Julien Lebon, François Baur, Patrick Lefevre, Damaris Ukeiwe, Julien Colot, Sylvie Tardieu, Cécile Cazorla.

**Methodology:** Julie Cagliero, François Baur, Patrick Lefevre, Emilie Follenfant, Cyrille Goarant, Damaris Ukeiwe, Julien Colot, Sylvie Tardieu, Catherine Werts, Cécile Cazorla.

**Project administration:** Julie Cagliero, Cyrille Goarant, Catherine Werts, Cécile Cazorla.

**Resources:** François Baur, Patrick Lefevre, Damaris Ukeiwe, Julien Colot, Sylvie Tardieu.

**Software:** Anne LOAREC.

**Supervision:** François Baur, Patrick Lefevre, Cyrille Goarant, Sylvie Tardieu, Catherine Werts, Cécile Cazorla.

**Writing – original draft:** Anne LOAREC, Catherine Werts.

**Writing – review & editing:** Julien Lebon, Patrick Lefevre, Cyrille Goarant, Julien Colot.

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
