## [Decision Letter · Decision Letter 0]

4 Jul 2025

PNTD-D-25-00843

Anti-inflammatory cytokine profile and Jarisch-Herxheimer reaction in Leptospirosis patients: a prospective case-series study in New Caledonia

Dear Dr. Werts,

Thank you for submitting your manuscript to PLOS Neglected Tropical Diseases. After careful consideration, we feel that it has merit but does not fully meet PLOS Neglected Tropical Diseases's publication criteria as it currently stands. Therefore, we invite you to submit a revised version of the manuscript that addresses the points raised during the review process.

Please submit your revised manuscript within 60 days Sep 02 2025 11:59PM. If you will need more time than this to complete your revisions, please reply to this message or contact the journal office at plosntds@plos.org. Please include the following items when submitting your revised manuscript:

We look forward to receiving your revised manuscript.

Kind regards,

Joseph M. Vinetz

Section Editor

Joseph Vinetz

Section Editor

Shaden Kamhawi

co-Editor-in-Chief

Paul Brindley

co-Editor-in-Chief

**Journal Requirements:**

3) Please upload a copy of Figure Figures 2 and 3 which you refer to in your text on page 8 and 16. Or, if the figure is no longer to be included as part of the submission please remove all reference to it within the text.

4) We notice that your supplementary Figures, and Tables are included in the manuscript file. Please remove them and upload them with the file type 'Supporting Information'. Please ensure that each Supporting Information file has a legend listed in the manuscript after the references list.

5) In the online submission form, you indicated that "Data sharing policy ensures that anonymised data secured inthe REDCAP sofware will be available upon request to interested researchers while addressing all security, legal, and ethical concerns. All readers may contact the corresponding authors to request the data.". All PLOS journals now require all data underlying the findings described in their manuscript to be freely available to other researchers, either

1. In a public repository

2. Within the manuscript itself

3. Uploaded as supplementary information.

**Reviewers' Comments:**

Reviewer's Responses to Questions

**Key Review Criteria Required for Acceptance?**

**Methods:**

-Are the objectives of the study clearly articulated with a clear testable hypothesis stated?

-Is the study design appropriate to address the stated objectives?

-Is the population clearly described and appropriate for the hypothesis being tested?

-Is the sample size sufficient to ensure adequate power to address the hypothesis being tested?

-Were correct statistical analysis used to support conclusions?

-Are there concerns about ethical or regulatory requirements being met?

Reviewer #1: JHR clinical score. The new JHR clinical scoring system is an important step forward and seems to have been validated by the various lab results presented in the paper. It would be helpful to understand the JHR score criteria more clearly. For example, how is fever defined (above what temperature)? How much of change in O2 saturation is considered a “decrease”? Do these changes need to happen within a particular time (eg 6 hours?) after treatment? How does the JHR score compare to JHR definitions used by others for syphilis or leptospirosis? Defining these criteria more clearly would be helpful for future studies by others.

The absence of measurable IL-1β is remarkable. Were positive controls included to make sure the assay was working? What is the sensitivity of the assay kit used to measure this cytokine? IL-1β is typically present in picogram-range concentrations and some detection methods may lack sensitivity. The authors should acknowledge IL-1β usually peaks early during an infection and may be compartmentalized during infection. Systemic IL-1β is rapidly cleared even as downstream cytokines like IL-6 and TNF remain elevated. IL-10 can act to suppress IL-1β production.

Reviewer #2: This is a well design study

1. Regarding to the definition of JHR, do we have the citation for this purpose?

2. The symptoms such as fever, chills, body ache and skin rash are often clinical manifestation of acute leptospirosis before treatment with an antibiotic, so worsening symptoms after treatment can be overlooked simply as signs of the underlying infection. How do we differentiate the finding from the nature of clinical manifestations?

**Results**

-Does the analysis presented match the analysis plan?

-Are the results clearly and completely presented?

-Are the figures (Tables, Images) of sufficient quality for clarity?

Reviewer #1: Clinical evolution. Of the 81 patients, apparently only 59 patients had JHR information recorded on the case report form. Please confirm whether the movement to ICU occurred during the observation period after treatment. Is it possible to correlate any of the time=0 laboratory results with subsequent movement to ICU? Movement to ICU apparently interfered with JHR information collection. At what timepoints after treatment did movement to ICU occur in the 11 patients? Is there any indication that movement to ICU was associated with any particular type of treatment, more severe JHR, or leptospiremia level? Related question: What was the range of JHR severity? Did movement to ICU or JHR occur more or less frequently in the patients who received a particular treatment modality such as escalating doses of amoxicillin and/or steroids? In other words, is there any indication of clinical benefit from the escalating doses of amoxicillin and/or steroid protocols? Answering this question could be a subject of a subsequent clinical trial.

Is one possible explanation for the lower liver and renal parameters in the JHR patients is that they were treated earlier in the course of infection than the patients without JHR. In other words, the patients without JHR had already cleared bacteremia because they were at a later stage of leptospirosis. Were any data collected from patients regarding the duration of time from onset of symptoms to treatment?

Reviewer #2: 1. What is the median day of fever at the time of enrollment? This timing may contribute to the level of cytokines at the presentation. How to reduct this lead time bias?

2. How do we explain the high rate of JHR in New Caledonia?

**Conclusions:**

-Are the conclusions supported by the data presented?

-Are the limitations of analysis clearly described?

-Do the authors discuss how these data can be helpful to advance our understanding of the topic under study?

-Is public health relevance addressed?

Reviewer #1: The authors should acknowledge that there are alternative explanations for the elevated IL-10 levels. IL-10 is an anti-inflammatory cytokine that can become elevated in a compensatory manner during severe infection. In the 2004 study of patients with septic shock by Monneret et al. (PMID: 15388260), IL-10 levels were correlated with severity and survival. In this JHR study, were IL-10 levels associated with more severe infection? How do the IL-10 levels observed in this JHR study compare to what has been observed in other studies of leptospirosis or other infectious diseases?

The authors are encouraged to explore the nexus between IL-10 immunosuppression, high levels of leptospiremia and JHR. The study results show higher IL-10 levels and higher levels of leptospiremia in the JHR patients than non-JHR patients. Based on these results, it seems plausible that IL-10 immunosuppression contributed to leptospiremia and that leptospiremia contributed to JHR.

Reviewer #2: I think the conclusion is a bit strong, for example conclusion as "This study confirms the stealth nature of leptospires, which do not induce inflammation but a potent anti-inflammatory response. It rejects the cytokine storm

hypothesis often cited in leptospirosis, and does not argue for the use of immunosuppressive

drugs. When we look to see the results, there is also increase level of IL-6, and TNF following time points, not only IL-10

**Editorial and Data Presentation Modifications?**

Reviewer #1: Including genotyping of infecting strains based on lfb-1 sequencing is potentially helpful. Although genotype did not seem to be linked to whether JHR occurred, it would be helpful to examine whether there is a link between genotype and clinical and/or laboratory severity. Specifically, was there any correlation between genotype and the level of leptospiremia?

Reviewer #2: (No Response)

**Summary and General Comments:**

Reviewer #1: This an innovative prospective study of 81 leptospirosis patients in New Caledonia compares the clinical and laboratory findings in leptospirosis patients with vs without Jarisch-Herxheimer reactions (JHR) following β-lactam antibiotic treatment. The novel findings of low baseline levels of pro-inflammatory cytokines and elevated IL-10 contribute greatly to our understanding of leptospirosis in that they challenge the cytokine storm model of leptospirosis.

Will follow-up studies be performed to examine the longer-term consequences of JHR in leptospirosis, such as post-treatment fatigue, organ dysfunction, or persistent cytokine elevation?

Reviewer #2: (No Response)

PLOS authors have the option to publish the peer review history of their article (what does this mean?). If published, this will include your full peer review and any attached files.

Reviewer #1: **Yes: **David Haake

Reviewer #2: No

**Figure resubmission:**
---

## [Decision Letter · Decision Letter 1]

5 Sep 2025

Dear Dr. Werts,

We are pleased to inform you that your manuscript 'Anti-inflammatory cytokine profile and Jarisch-Herxheimer reaction in Leptospirosis patients: a prospective case-series study in New Caledonia' has been provisionally accepted for publication in PLOS Neglected Tropical Diseases.

Best regards,

Joseph M. Vinetz

Section Editor

Joseph Vinetz

Section Editor

Shaden Kamhawi

co-Editor-in-Chief

Paul Brindley

co-Editor-in-Chief

Reviewer #1:

Reviewer #2:

Reviewer's Responses to Questions

**Key Review Criteria Required for Acceptance?**

**Methods**

-Are the objectives of the study clearly articulated with a clear testable hypothesis stated?

-Is the study design appropriate to address the stated objectives?

-Is the population clearly described and appropriate for the hypothesis being tested?

-Is the sample size sufficient to ensure adequate power to address the hypothesis being tested?

-Were correct statistical analysis used to support conclusions?

-Are there concerns about ethical or regulatory requirements being met?

Reviewer #1: No additional comments.

Reviewer #2: (No Response)

**Results**

-Does the analysis presented match the analysis plan?

-Are the results clearly and completely presented?

-Are the figures (Tables, Images) of sufficient quality for clarity?

Reviewer #1: No additional comments.

Reviewer #2: (No Response)

**Conclusions**

-Are the conclusions supported by the data presented?

-Are the limitations of analysis clearly described?

-Do the authors discuss how these data can be helpful to advance our understanding of the topic under study?

-Is public health relevance addressed?

Reviewer #1: No additional comments.

Reviewer #2: (No Response)

**Editorial and Data Presentation Modifications?**

Reviewer #1: No additional comments.

Reviewer #2: none

**Summary and General Comments**

Reviewer #1: The authors have satisfactorily addressed the comments that I provided in the initial review of the manuscript.

Reviewer #2: The authors adequately response to my questions, and I do not have any further questions.

PLOS authors have the option to publish the peer review history of their article (what does this mean?). If published, this will include your full peer review and any attached files.

Reviewer #1: No

Reviewer #2: No

---

## [Editor Report · Acceptance letter]

Dear Dr. Werts,

We are delighted to inform you that your manuscript, "Anti-inflammatory cytokine profile and Jarisch-Herxheimer reaction in Leptospirosis patients: a prospective case-series study in New Caledonia," has been formally accepted for publication in PLOS Neglected Tropical Diseases.

Best regards,

Shaden Kamhawi

co-Editor-in-Chief

Paul Brindley

co-Editor-in-Chief
